# FORWARD PREDICTION FOR PHYSICAL REASONING

## ABSTRACT

Physical reasoning requires forward prediction: the ability to forecast what will happen next given some initial world state. We study the performance of state-of-the-art forward-prediction models in the complex physical-reasoning tasks of the PHYRE benchmark (Bakhtin et al., 2019). We do so by incorporating models that operate on object or pixel-based representations of the world into simple physical-reasoning agents. We find that forward-prediction models can improve physical-reasoning performance, particularly on complex tasks that involve many objects. However, we also find that these improvements are contingent on the test tasks being small variations of train tasks, and that generalization to completely new task templates is challenging. Surprisingly, we observe that forward predictors with better pixel accuracy do not necessarily lead to better physical-reasoning performance. Nevertheless, our best models set a new state-of-the-art on the PHYRE benchmark.

## 1 INTRODUCTION

When presented with a picture of a Rube Goldberg machine, we can predict how the machine works. We do so by using our intuitive understanding of concepts such as force, mass, energy, collisions, *etc.*, to *imagine* how the machine state would evolve once released. This ability allows us to solve real world physical-reasoning tasks, such as how to hit a billiards cue such that the ball ends up in the pocket, or how to balance the weight of two children on a see-saw. In contrast, physical-reasoning abilities of machine-learning models have largely been limited to closed domains such as predicting dynamics of multi-body gravitational systems (Battaglia et al., 2016), stability of block towers (Lerer et al., 2016), or physical plausibility of observed dynamics (Riochet et al., 2018). In this work, we explore the use of *imaginative*, forward-prediction approaches to solve complex physical-reasoning puzzles. We study modern object-based (Battaglia et al., 2016; Sanchez-Gonzalez et al., 2020; Watters et al., 2017) and pixel-based (Finn et al., 2016; Ye et al., 2019; Hafner et al., 2020) forward-prediction models in simple search-based agents on the PHYRE benchmark (Bakhtin et al., 2019). PHYRE tasks involve placing one or two balls in a 2D world, such that the world reaches a state with a particular property (*e.g.*, two balls are touching) after being played forward. PHYRE tasks are very challenging because small changes in the action (or the world) can have a very large effect on the efficacy of an action; see Figure 1 for an example. Moreover, PHYRE tests models' ability to generalize to *completely* new physical environments at test time, a significantly harder task than prior work that mostly varies number or properties of objects in the same environment. As a result, physical-reasoning agents may struggle even when their forward-prediction model works well.

Nevertheless, our best agents substantially outperform the prior state-of-the-art on PHYRE. Specifically, we find that forward-prediction models can improve the performance of physical-reasoning agents when the models are trained on tasks that are very similar to the tasks that need to be solved at test time. However, we find forward-prediction based agents struggle to generalize to truly unseen tasks, presumably, because small deviations in forward predictions tend to compound over time. We also observe that better forward prediction does not always lead to better physical-reasoning performance on PHYRE (*c.f.* Buesing et al. (2018) for similar observations in RL). In particular, we find that object-based forward-prediction models make more accurate forward predictions but pixel-based models are more helpful in physical reasoning. This observation may be the result of two key advantages of models using pixel-based state representations. First, it is easier to determine whether a task is solved in a pixel-based representation than in an object-based one, in fully observable 2D environments like PHYRE. Second, pixel-based models facilitate end-to-end training of the

forward-prediction model and the task-solution model in a way that object-based models do not in the absence of a differentiable renderer (Liu et al., 2019; Loper & Black, 2014).

## 2 RELATED WORK

Our study builds on a large body of prior research on forward prediction and physical reasoning. We discuss most closely related work in this section and report additional prior work in Appendix B.

**Forward prediction** models attempt to predict the future state of objects in the world based on observations of past states. Such models operate either on *object-based* (proprioceptive) representations or on *pixel-based* state representations. A popular class of *object-based* models use graph neural networks to model interactions between objects (Kipf et al., 2018; Battaglia et al., 2016), for example, to simulate environments with thousands of particles (Sanchez-Gonzalez et al., 2020; Li et al., 2019). Another class of object-based models explicitly represents the Hamiltonian or Lagrangian of the physical system (Greydanus et al., 2019; Cranmer et al., 2020; Chen et al., 2019). While promising, such models are currently limited to simple point objects and physical systems that conserve energy. Hence, they cannot currently be used on PHYRE, which contains dissipative forces and extended objects. Modern *pixel-based* forward-prediction models extract state representations by applying a convolutional network on the observed frame(s) (Watters et al., 2017; Kipf et al., 2020) or on object segments (Ye et al., 2019; Janner et al., 2019). The models perform forward prediction on the resulting state representation using graph neural networks (Kipf et al., 2020; Ye et al., 2019; Li et al., 2020), recurrent neural networks (Xingjian et al., 2015; Hochreiter & Schmidhuber, 1997; Cho et al., 2014; Finn et al., 2016), or a physics engine (Wu et al., 2017) . The models can be trained to predict object state (Watters et al., 2017), perform pixel reconstruction (Villegas et al., 2017; Ye et al., 2019), transform the previous frames (Ye et al., 2018; 2019; Finn et al., 2016), or produce a contrastive state representation (Kipf et al., 2020; Hafner et al., 2020).

**Physical reasoning** tasks gauge a system's ability to intuitively reason about physical phenomena (Battaglia et al., 2013; Kubricht et al., 2017). Prior work has developed models that predict whether physical structures are stable (Lerer et al., 2016; Groth et al., 2018; Li et al., 2016), predict whether physical phenomena are plausible (Riochet et al., 2018), describe or answer questions about physical systems (Yi et al., 2020; Rajani et al., 2020), perform counterfactual prediction in physical worlds (Baradel et al., 2020), predict effect of forces (Mottaghi et al., 2016; Wang et al., 2018), or solve physical puzzles/games (Allen et al., 2020; Bakhtin et al., 2019; Du & Narasimhan, 2019). Unlike other physical reasoning tasks, physical-puzzle benchmarks such as PHYRE (Bakhtin et al., 2019) and Tools (Allen et al., 2020) incorporate a full physics simulator, and contain a large set of physical environments to study generalization. This makes them particularly suitable for studying the effectiveness of forward prediction for physical reasoning, and we adopt the PHYRE benchmark in our study for that reason.

**Inferring object representations** involve techniques like generative models and attention mechanisms to decompose scenes into objects (Eslami et al., 2016; Greff et al., 2019; Burgess et al., 2019; Engelcke et al., 2019). Many techniques also leverage the motion information for better decomposition or to implicitly learn object dynamics (Kipf et al., 2020; van Steenkiste et al., 2018; Crawford & Pineau, 2020; Kosiorek et al., 2018). While relevant to our exploration of pixel-based methods as well, we leverage the simplicity of PHYRE visual world to extract object-like representations simply using connected component algorithm in our approaches (*c.f.* STN in Section 4.1). However, more sophisticated approaches could help further improve the performance, and would be especially useful for more visually complex and 3D environments.

## 3 PHYRE BENCHMARK

In PHYRE, each *task* consists of an *initial state* that is a $256 \times 256$ image. Colors indicate object properties; for instance, black objects are static while gray objects are dynamic and neither are involved in the goal state. PHYRE defines two task *tiers* (B and 2B) that differ in their action space. An *action* involves placing one ball (in the B tier) or two balls (in the 2B tier) in the image. Balls are parameterized by their position and radius, which determine the ball's mass. An action solves the task if the blue or purple object touches the green object (the *goal state*) for a minimum of three seconds

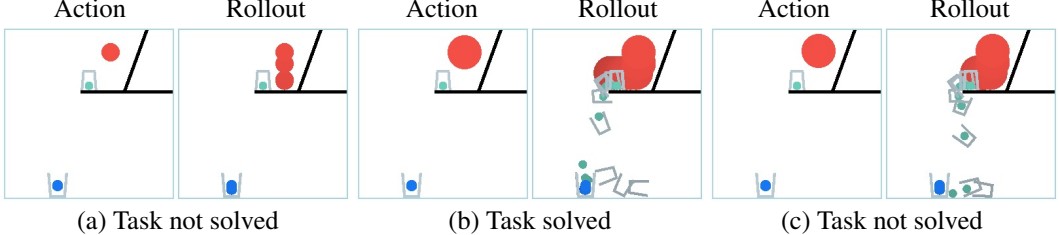

| Action | Rollout | Action | Rollout | Action | Rollout |

(a) Task not solved    (b) Task solved    (c) Task not solved

Figure 1: PHYRE tasks require placing an object (the red ball) in the scene, such that when the simulation is rolled out, the blue and green objects touch for at least three seconds. In (a), the ball is too small and does not knock the green ball off the platform. In (b), the ball is larger and solves the task. In (c), the ball is placed slightly farther left, which results in the task not being solved. Small variations in the selected action (or the scene) can have a large effect on the efficacy of the action.

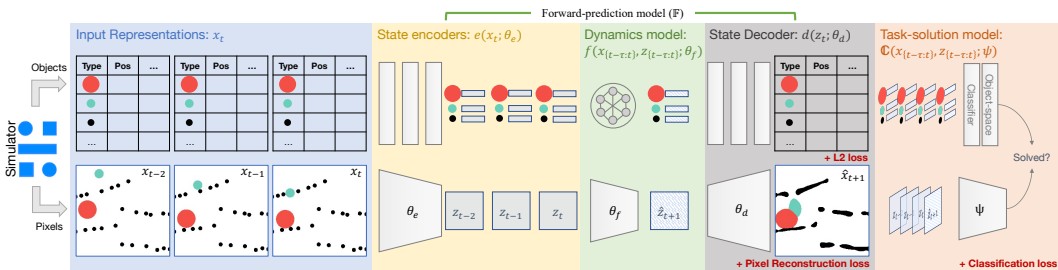

Figure 2: We study models that take as input a set of initial state via an object-based or a pixel-based representation (blue box). We input the representation into a range of forward-prediction models, which generally comprise an *encoder* (yellow box), a *dynamics model* (green box), and a *decoder* (gray box). We feed that output to a *task-solution model* (red box) that predicts whether the goal state is reached. At inference time, we search over actions that alter the initial state by adding additional objects to the state. For each action (and corresponding initial state) we predict a task-solution probability; we then select the action most likely to solve the task.

when the simulation is rolled out. Figure 1 illustrates the challenging nature of PHYRE tasks: small variations can change incorrect actions (Figure 1(a) and (c)) into a correct solution (Figure 1(b)).

Each tier in PHYRE contains 25 *task templates*. A task template contains 100 tasks that are structurally similar but that differ in the initial positions of the objects. Performance on PHYRE is measured in two settings. The *within-template* setting defines a train-test split over tasks, such that training and test tasks can contain different instantiations of the same template. The *cross-template* setting splits across templates, such that training and test tasks never correspond to the same template. A PHYRE *agent* can make multiple *attempts* at solving a task. The performance of the agent is measured by the *area under the success curve* (AUCCESS; (Bakhtin et al., 2019)), which ranges from 0 to 100 and is higher when the agent needs fewer attempts to solve a task. Performance is averaged over 10 random splits of tasks or templates. In addition to AUCCESS, we also measure a *forward-prediction accuracy* (FPA) that does not consider whether an action solves a task. We define FPA as the percentage of pixels that match the ground-truth in a 10-second *rollout* at 1 frame per second (fps); we only consider pixels that correspond to *dynamic* objects when computing forward-prediction accuracy. Please refer to Appendix E for exact implementation details.

## 4    METHODS

We develop physical-reasoning agents for PHYRE that use learned *forward-prediction model*s in a *search strategy* to find actions that solve a task. The search strategy maximizes the score of a *task-solution model* that, given a world state, predicts whether that state will lead to a task solution. Figure 2 illustrates how our forward-prediction and task-solution models are combined. We describe both types of models, as well as the search strategy we use, separately below.

### 4.1 FORWARD-PREDICTION MODELS

At time $t$, a forward-prediction model $\mathbb{F}$ aims to predict the next state, $\hat{x}_{t+1}$, of a physical system based on a series of $\tau$ past states of that system. $\mathbb{F}$ consists of a state encoder $e$, a forward-dynamics model $f$, and a state decoder $d$. The past states $\{x_{t-\tau}, \ldots, x_t\}$ are first encoded into latent representations $\{z_{t-\tau}, \ldots, z_t\}$ using a learned encoder $e$ with parameters $\theta_e$, *i.e.*, $z_t = e(x_t; \theta_e)$. The latent representations are then passed into the forward-dynamics model $f$ with parameters $\theta_f$: $f(\{x_{t-\tau}, \ldots, x_t\}, \{z_{t-\tau}, \ldots, z_t\}; \theta_f) \to \hat{z}_{t+1}$. Finally, the predicted future latent representation is decoded using the decoder $d$ with parameters $\theta_d$: $d(\hat{z}_{t+1}; \theta_d) \to \hat{x}_{t+1}$ . We learn the model parameters $\Theta = (\theta_e, \theta_f, \theta_d)$ on a large training set of observations of the system's dynamics. We experiment with forward-prediction models that use either object-based or pixel-based state representations.

**Object-based models.** We experiment with two object-based forward-prediction models that capture interactions between objects: interaction networks (Battaglia et al., 2016) and transformers (Vaswani et al., 2017). Both object-based forward-prediction models represent the system's state as a set of tuples that contain object type (ball, stick, *etc.*), location, size, color, and orientation. The models are trained by minimizing the mean squared error between the predicted and observed state.

- **Interaction networks (`IN`;** Battaglia et al. (2016)**)** maintain a vector representation for each object in the system at time $t$. Each vector captures information about the object's type, position, and velocity. A relationship is computed for each ordered pair of objects, designating the first object as the sender and the second as the receiver of the relation. The relation is characterized by the concatenation of the two objects' feature vectors and a one-hot encoding representing the sender object's attribute of static or dynamic. The dynamics model embeds the relations into "effects" per object using a multilayer perceptron (MLP). The effects exerted on an object are summed into a single effect per object. This aggregated effect is concatenated with the object's previous state vector, from a particular temporal offset, along with a placeholder for external effects, *e.g.*, gravity. The result is passed through another MLP to predict velocity of the object. We use two interaction networks with different temporal offsets (Watters et al., 2017), and aggregate the results in an MLP to generate the final velocity prediction. The decoder then sums the object's predicted velocity with the previous position to obtain the new position of the object.
- **Transformers (`Tx`;** Vaswani et al. (2017)**)** also maintain a representation per object: they encode each object's state using a 2-layer MLP. In contrast to `IN`, the dynamics model $f$ in `Tx` is a Transformer that uses self-attention layers over the latent representation to predict the future state. We add a sinusoidal temporal position encoding (Vaswani et al., 2017) of time $t$ to the features of each object. The resulting representation is fed into a Transformer encoder with 6 layers and 8 heads. The output representation is decoded using a MLP and added to the previous state to obtain the future state prediction.

**Pixel-based models.** In contrast to object-based models, pixel-based forward-prediction models do not assume direct access to the attribute values of the objects. Instead, they operate on images depicting the object configuration, and maintain a single, global world state that is extracted by an image encoder. Our image encoder $e$ is a ResNet-18 network (He et al., 2016) that is clipped at the `res4` block. Objects in PHYRE can have seven different colors; hence, the input of the network consists of seven channels with binary values that indicate object presence, consistent with prior work (Bakhtin et al., 2019). The representations extracted from the past $\tau$ frames are concatenated before being input into the two models we study.

- **Spatial transformer networks (`STN`;** (Jaderberg et al., 2015)**)** split the input frame into segments by detecting objects (Ye et al., 2019), and then encode each object using the encoder $e$. Specifically, we use a simple connected components algorithm (Weaver, 1985) to split each frame channel into object segments. The dynamics model concatenates the object channels for the $\tau$ input frames, and predicts a rotation and translation for each channel corresponding to the last frame using a small convolutional network. The decoder applies the predicted transformation to each channel. The resulting channels are combined into a single frame prediction by summing them. Inspired by modern keypoint localizers (He et al., 2017), we train `STN`s by minimizing the *spatial cross-entropy*, which sums the cross-entropies of $H \times W$ softmax predictions over all seven channels.
- **Deconvolutional networks (`Dec`)** directly predict the pixels in the next frame using a deconvolutional network that does not rely on a segmentation of the input frame(s). The representations for the last $\tau$ frames are concatenated along the channel dimension, and passed through a small convolutional network to generate a latent representation for the $t + 1^{th}$ frame. Latent represen-

tation $\hat{z}_{t+1}$ is then decoded to pixels using a deconvolutional network, implemented as series of five transposed-convolution and (bilinear) upsampling layers, with intermediate ReLU activation functions. We found `Decs` are best trained by minimizing the *per-pixel cross-entropy*, which sums the cross-entropy of seven-way softmax predictions at each pixel.

## 4.2 TASK-SOLUTION MODELS

We use our forward-prediction models in combination with a *task-solution model* that predicts whether a rollout solves a physical-reasoning task. In the physical-reasoning tasks we consider, the task-solution model needs to recognize whether two particular target objects are touching (task solved) or not (task not solved). This recognition is harder than it seems, particularly when using an object-based state representation. For example, evaluating whether or not the centers of two balls are "near" is insufficient because the radiuses of the balls need to be considered as well. For more complex objects, the model needs to evaluate complex relations between the two objects, as well as recognize other objects in the scene that may block contact. We note that good task-solution models may also correct for errors made by the forward-prediction model.

Per Figure 2, we implement the task solution model using a simple binary classifier $\mathbb{C}$ with parameters $\psi$. The classifier receives the $\tau + \tau'$ (initial and predicted) frames and/or latent representations as input from the forward-prediction model. We provide the input frames as well to account for potentially poor performance of the dynamics model on certain tasks; in those cases the task-solution model can learn to ignore the rollout and only use input frames to make a prediction. It then produces a binary prediction: $\mathbb{C}(x_0, \ldots, x_\tau, z_0, \ldots, z_\tau, \hat{x}_{\tau+1}, \ldots, \hat{x}_{\tau+\tau'}, \hat{z}_{\tau+1}, \ldots, \hat{z}_{\tau+\tau'}; \psi) \rightarrow [-1, +1]$. Because both types of forward-prediction models produce different outputs, we experiment with object-based classifiers and pixel-based classifiers that make predictions based on simulation state represented by object features or pixels respectively. We also experiment with pixel-based classifiers on object-based forward-prediction models by rendering the object-based state to pixels first.

- **Object-based classifier (`Tx-Cls`).** We use a Transformer (Vaswani et al., 2017) model that encodes the object type and position into a 128-dimensional encoding using a two-layer MLP. As before, a sinusoidal temporal position encoding is added to each object's features. The resulting encodings for all objects over the $\tau + \tau'$ time steps are concatenated, and used in a 16-head, 8-layer transformer encoder with LayerNorm. The resulting representations are input into another MLP that performs a binary classification that predicts whether or not the task is solved.
- **Pixel-based classifier (`Conv3D-{Latent,Pixel}`).** Our pixel-based classifier poses the problem of classifying task solutions as a video-classification problem. Specifically, we adopt a 3D convolutional network for video classification (Tran et al., 2015; 2018; Carreira & Zisserman, 2017). We experiment with two variants of this model: (1) `Conv3D-Latent`: the latent state representations $(z, \hat{z})$ are concatenated along the temporal dimension, and passed through a stack of 3D convolutions with intermediate ReLUs followed by a linear classifier; and (2) `Conv3D-Pixel`: the pixel representations $(x, \hat{x})$ are encoded using a ResNet-18 (up to `res4`), and classifications are made by the `Conv3D-Latent` model. `Conv3D-Pixel` can also be used in combination with object-based forward-prediction models, as the predictions of those models can be rendered.

## 4.3 SEARCH STRATEGY

We compose a forward-prediction model $\mathbb{F}$ and a task-solution model $\mathbb{C}$ to form a scoring function for action proposals. An action adds one or more additional objects to the initial world state. We sample $K$ actions uniformly at random and evaluate the value of the scoring function for the sampled actions. To evaluate the scoring function, we alter the initial state with the action, use the resulting state as input into the forward-prediction model, and evaluate the task-solution model on the output of the forward-prediction model. The search strategy selects the action that is most likely to solve the task according to the task-solution model, based on the output of the forward-prediction model.

## 5 EXPERIMENTS

We evaluate the performance of the forward-prediction models on the B-tier of the challenging PHYRE benchmark (Bakhtin et al. (2019); see Figure 1). We present our experimental setup and the results of our experiments below. We provide trained models and code reproducing our results online.

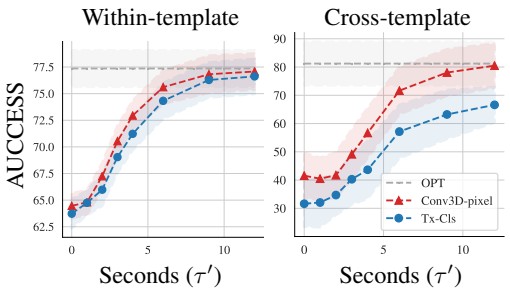 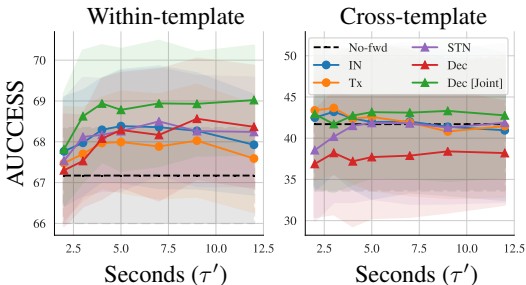

Figure 3: AUCCESS of object-based (●) and pixel-based (▲) task-solution model ($y$-axis) applied on state obtained by rolling out an *oracle* forward-prediction model for $\tau'$ seconds ($x$-axis). AUCCESS of the OPTIMAL agent is shown for reference. Shaded regions indicate standard deviation across 10 folds. We note that object-based task-solution models struggle compared to pixel-based ones, especially in cross-template settings.

Figure 4: AUCCESS of task-solution model applied on state obtained by rolling out five object-based (●) and pixel-based (▲) forward-prediction models ($y$-axis) for $\tau'$ seconds ($x$-axis). Forward-prediction models were initialized with $\tau = 3$ ground-truth states. AUCCESS of agent without forward prediction (No-fwd) is shown for reference. Results are presented for the within-template (left) and cross-template (right) settings.

## 5.1 EXPERIMENTAL SETUP

**Training.** To generate training data for our models, we sample task-action pairs in a balanced way: half of the samples solve the task and the other half do not. We generate training examples for the forward-prediction models by obtaining frames from the simulator at 1 fps, and sampling $\tau$ consecutive frames used to bootstrap the forward model from a random starting point in this obtained rollout. The model is trained to predict frames that succeed the selected $\tau$ frames. For the task-solution model, we always sample $\tau$ frames from the starting point of the rollout, or frame 0. Along with these $\tau$ frames, the task-solution model also gets the $\tau'$ autoregressively predicted frames from the forward-prediction model as input. We use $\tau = 3$ for most experiments, and eventually relax this constraint to use $\tau = 1$ frame when comparing to the state-of-the-art in the next section.

We train most forward-prediction models using *teacher forcing* (Williams & Zipser, 1989): we only use ground-truth states as input into the forward model during training. The only exception is Dec, for which we observed better performance when predicted states are used as input when training. Furthermore, since Dec is trainable without teacher forcing, we are able to train it jointly with the task-solution model, as it no longer requires picking a random point in the rollout to train the forward model. In this case, we train both models from frame 0 of each simulation with equal weights on both losses, and refer to this model as Dec [Joint]. For object-based models, we add a small amount of Gaussian noise to object states during training to make the model robust (Battaglia et al., 2016). We train all task-solution and pixel-based forward-prediction models using mini-batch SGD, and train object-based forward-prediction models with Adam. We selected hyperparameters for each model based on the AUCCESS on the first fold in the within-template setting; see Appendix I.

**Evaluation.** At inference time, we bootstrap the forward-prediction models with $\tau$ initial ground-truth states from the simulator for a given action, and autoregressively predict $\tau'$ future states. The $\tau + \tau'$ states are then passed into the task-solution model to predict whether the task will be solved or not by this action. Following (Bakhtin et al., 2019), we use the task-solution model to score a fixed set of $K = 1,000$ (unless otherwise specified) randomly selected actions for each task. We rank these actions based on the task-solution model score to measure AUCCESS. We also measure forward-prediction accuracy (FPA; see Section 3) on the validation tasks for 10 random actions each, half of which solve the task and the other half that do not. Following (Bakhtin et al., 2019), we repeat all experiments for 10 folds and report the mean and standard deviation of the AUCCESS and FPA.

## 5.2 RESULTS

We organize our experimental results based on a series of research questions.

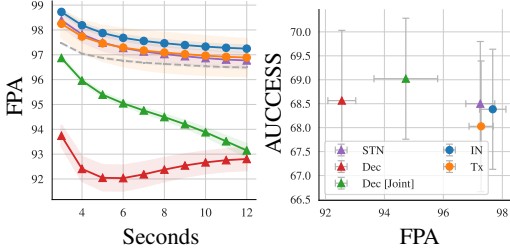

Table 1: AUCCESS and success percentage of our Dec [Joint] agents using $\tau' = 0$ (no roll-out, frame-level model) and $\tau' = 10$ (full roll-out) compared to current state-of-the-art agents (Bakhtin et al., 2019) on the PHYRE benchmark. In contrast to prior experiments, all agents here are conditioned on $\tau = 1$ initial frame. Our agents outperform all prior work on both settings and metrics of the PHYRE benchmark.

Figure 5: **Left:** Within-template forward-prediction accuracy (FPA) after $\tau'$ seconds roll out of five forward-prediction models, and an "identity" baseline that repeats the last input frame (dashed gray line). **Right:** Maximum AUCCESS value across roll-out as a function of forward-prediction accuracy averaged over $\tau' = 10$ seconds for the five models. Shaded regions and error bars indicate standard deviation over 10 folds.

| | AUCCESS | | Success %age | |
|---|---|---|---|---|
| | **Within** | **Cross** | **Within** | **Cross** |
| RAND (Bakhtin et al., 2019) | $13.7_{\pm 0.5}$ | $13.0_{\pm 5.0}$ | $7.7_{\pm 0.8}$ | $6.8_{\pm 5.0}$ |
| MEM (Bakhtin et al., 2019) | $2.4_{\pm 0.3}$ | $18.5_{\pm 5.1}$ | $2.7_{\pm 0.5}$ | $15.2_{\pm 5.9}$ |
| DQN (Bakhtin et al., 2019) | $77.6_{\pm 1.1}$ | $36.8_{\pm 9.7}$ | $81.4_{\pm 1.9}$ | $34.5_{\pm 10.2}$ |
| Ours ($\tau' = 0$) | $76.7_{\pm 0.9}$ | $\mathbf{40.7_{\pm 7.7}}$ | $80.7_{\pm 1.5}$ | $\mathbf{40.1_{\pm 8.2}}$ |
| Ours ($\tau' = 10$) | $\mathbf{80.0_{\pm 1.2}}$ | $40.3_{\pm 8.0}$ | $\mathbf{84.1_{\pm 1.8}}$ | $39.2_{\pm 8.6}$ |

**Can perfect forward prediction solve PHYRE physical reasoning?** We first evaluate if perfect forward-prediction can solve physical reasoning on PHYRE. We do so by using the PHYRE simulator as the forward-prediction model, and applying task-solution models on the predicted state. We exclude the Conv3D-Latent task-solution model in this experiment because it requires the latent representation of learned forward-prediction model, which the simulator can not provide. Figure 3 shows the AUCCESS of these models as a function of the number of seconds the forward-prediction model is rolled out. We compare model performance with that of the OPTIMAL agent (Bakhtin et al., 2019), which is an agent that achieves the maximum attainable performance given that we rank only $K$ actions. We observe that task-solution models work nearly perfectly in the within-template setting when the forward-prediction is rolled out for $\tau' \approx 10$ seconds. We also observe that pixel-based task-solution models outperform object-based models, especially in the cross-template setting. This suggests that it is more difficult for object-based models to determine whether or not two objects are touching than for pixel-based models, presumably, because the computations required are more complex. In preliminary experiments, we found that Conv3D-Latent outperforms Conv3D-Pixel when combined with learned pixel-based forward-prediction models (see Appendix D.1). Therefore, we use Conv3D-Latent as the task-solution model for pixel-based models and Conv3D-Pixel for object-based models (by rendering object-based predictions) in the remaining experiments.

**How well do forward-prediction models solve PHYRE physical reasoning?** We evaluate performance of our learned forward-prediction models on the PHYRE tasks. Akin to the previous experiment, we roll out the forward-prediction model for $\tau'$ seconds and evaluate the corresponding task-solution model on the $\tau'$ state predictions and the $\tau = 3$ input states. Figure 4 presents the AUCCESS of this approach as a function of the number of seconds ($\tau'$) that the forward-prediction models were rolled out. The AUCCESS of an agent without forward prediction (No-fwd) is shown for reference. The results show that forward prediction can improve AUCCESS by up to $2\%$ in the within-template setting. The pixel-based Dec model performs similarly to models that operate on object-based states, either extracted (STN) or ground truth (IN and Tx). Furthermore, Dec allows for end-to-end training of the forward-prediction and the pixel-based task-solution models. The resulting Dec [Joint] model performs the best in our experiments, which is why we focus on it in subsequent experiments. Similar joint training of object-based models (*c.f.* Appendix D.2) yields smaller improvements due to limitations of object-based task-solution models. Although the within-template results suggest that forward prediction can help physical reasoning, AUCCESS plateaus after $\tau' \approx 5$ seconds. This suggests that forward-prediction models are only truly accurate on PHYRE for a short period of time. Also, forward-prediction models help little in the cross-template setting, suggesting limited generalization across templates.

**Does better forward-prediction imply better PHYRE physical reasoning?** Figure 5 (left) measures the forward-prediction accuracy (FPA) of our forward-prediction models after $\tau'$ seconds of rolling out the models. We observe that FPA generally decreases with roll-out time although,

Figure 6: **Left:** Per-template AUCCESS of `Dec [Joint] 1f` with $\tau' = 0$ (no forward prediction) and $\tau' = 10$ (forward prediction) of five task templates that benefit the least from forward prediction (left) and five templates that benefit the most (right). **Right:** Per-template AUCCESS of the $\tau' = 0$ model as a function of the number of objects in the task template (left) and improvement in per-template AUCCESS, called $\Delta$ AUCCESS, obtained by the $\tau' = 10$ model (right).

interestingly, `Dec` recovers over time (*c.f.* Appendix C). While all models obtain a fairly high FPA, models that utilize object-centric representations (`IN`, `Tx`, and `STN`) clearly outperform their pixel-based counterparts. This is intriguing because, in prior experiments, `Dec` models performed best on PHYRE in terms of AUCCESS. To investigate this in more detail, Figure 5 (right) shows FPA averaged over 10 seconds as a function of the maximum AUCCESS over that time. The results confirm that more pixel-accurate forward predictions do *not* necessarily increase performance on PHYRE's physical-reasoning tasks.

**How do forward-prediction agents compare to the state-of-the-art on PHYRE?** Hitherto, all our experiments assumed access to $\tau = 3$ input frames, which is not the setting considered by Bakhtin et al. (2019). To facilitate comparisons with prior work, we develop an `Dec` agent that requires only $\tau = 1$ input frame: we pad the first frame with 2 empty frames and train the model exclusively on roll-outs that start from the first frame and do not use teacher forcing. We refer to the resulting model as `Dec [Joint] 1f`. Table 1 compares the performance of `Dec [Joint] 1f` to the state-of-the-art on PHYRE in terms of AUCCESS and success percentage @ 10 (*i.e.*, the percentage of tasks that were solved within 10 attempts). The results show that `Dec [Joint] 1f` outperforms the previous best reported agents in terms of metrics in both the within and the cross-template settings. In the within-template setting, the performance of `Dec [Joint] 1f` increases substantially for large $\tau'$. This demonstrates the benefits of using a forward-prediction modeling approach to PHYRE in that setting. Having said that, forward-prediction did not help in the cross-template setting, presumably, because rollouts on unseen templates, while realistic, were not accurate enough to solve the tasks.

**Which PHYRE templates benefit from using a forward-prediction model?** To investigate this, we compare `Dec [Joint] 1f` at $\tau = 0$ (*i.e.*, no forward-prediction) and $\tau = 10$ seconds in terms of per-template AUCCESS. We define per-template AUCCESS as the average AUCCESS over all tasks in a template in the within-template setting. Figure 6 (left) shows the per-template AUCCESS for the five templates in which forward-prediction models help the least (left five groups) and the five templates in which these models help the most (right five). Qualitatively, we observe that forward prediction does not help much in "simple" tasks that comprise a few objects, whereas it helps a lot in more "complex" tasks. This is corroborated by the results in Figure 6 (right), in which we show AUCCESS and the improvement in AUCCESS due to forward modeling ($\Delta$ AUCCESS) as a function of the number of objects in the task. We observe that AUCCESS decreases ($\rho = -0.4$) with the number of objects in the task, but that the benefits of forward predictions increase ($\rho = 0.6$).

# 6 DISCUSSION

While the results of our experiments demonstrate the potential of forward prediction for physical reasoning, they also highlight that much work is still needed for the full potential to materialize. The main challenge is that physical environments such as PHYRE are inherently chaotic: a small change in an action (or scene) may drastically affect the action's efficacy. Figure 7 shows an example of this: our models produce realistic forward predictions (also see Appendix A), but they may still select incorrect actions. Furthermore, PHYRE expects a single model to learn all templates and generalize across templates, exacerbating this challenge. Notably, recent work on particle-based representations (Li et al., 2019; Sanchez-Gonzalez et al., 2020) has shown successful initial results

| Input | Simulator | STN | Input | Simulator | STN | Input | Simulator | STN |

Action (red ball) solving the task.        Using a slightly smaller ball.        Using a slightly larger ball.

Figure 7: Rollouts for three slightly different actions on the same task by: (1) the simulator and (2) a STN trained only on tasks from the corresponding task template. Although the STN produces realistic rollouts, its predictions do not perfectly match the simulator. The small variations in action change whether the action solves the task. The STN model is unable to capture those variations effectively.

in terms of generalization by limiting the set of object types to only one (*viz.*, particles). However, such approaches are yet to be studied in extreme generalization settings as expected in PHYRE, and are constrained in accuracy by the underlying particle representation of extended objects, i.e. a pixel level decomposition would be most accurate though computationally infeasible. Nevertheless, we believe this is an interesting direction and we plan to study it in the context of PHYRE in future work. Furthermore, much of our analysis is specific to 2D environments, as used in the physical challenge benchmarks like TOOLS (Allen et al., 2020) and PHYRE. Extending this analysis to 3D or partially observable environments would also be interesting future work, where object-based models may have an advantage over pixel-based. Finally, experimenting with more sophisticated scene-decomposition approaches (Kosiorek et al., 2018; Crawford & Pineau, 2020; van Steenkiste et al., 2018), can allow for better joint training of object-centric approaches and further improve performance on the PHYRE tasks.

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

## A    ROLLOUT VISUALIZATIONS

We first show rollouts when our model is trained on train tasks from a single template, and evaluated from the val tasks on that template. We chose one of the hardest templates in terms of FPA, Template 18. This evaluation is comparable to most prior work (Battaglia et al., 2016; Sanchez-Gonzalez et al., 2020), where variations of single environment are used during training and testing. Note that our model's rollouts are quite high fidelity in this case, as expected.

- **Within-Template Template 18 (fold 0):**
  http://fwd-pred-phyre.s3-website.us-east-2.amazonaws.com/
  within-temp18only/

However, PHYRE requires training a single model for all the templates in the dataset. We now show a comprehensive visualization of rollouts for all our forward-prediction models (as were used to compute forward-prediction accuracy), for both within and cross-template settings. Note that the rollout fidelity drops, understandably as the model needs to capture all the different templates.

- **Within-Template (fold 0):**
  http://fwd-pred-phyre.s3-website.us-east-2.amazonaws.com/within/
- **Cross-Template (fold 0):**
  http://fwd-pred-phyre.s3-website.us-east-2.amazonaws.com/cross/

## B    ADDITIONAL RELATED WORK

Due to limited space in the main paper, we discuss some additional related work in this section.

**Additional work in Forward Prediction.** Learning of forward predictive models, including using neural networks, has a long history (Grzeszczuk et al., 1998; Byravan & Fox, 2017). As discussed in the main paper, many recent approaches tend to factorize input into object or particle level representations, and learns compositional models for forward prediction. Additional work in that direction includes Fragkiadaki et al. (2016); Chang et al. (2017); Mrowca et al. (2018); Veerapaneni et al. (2020).

**Video Prediction.** Also related is prior work in conditional pixel generation or video prediction, as that typically requires an implicit understanding of physics. Popular approaches model the past frames using a variant of recurrent neural network (Xingjian et al., 2015; Hochreiter & Schmidhuber, 1997; Cho et al., 2014) and make predictions directly using a decoder (Villegas et al., 2017), or as a transformation of the previous frames using optical flow (Ye et al., 2018) or spatial transformations (Ye et al., 2019; Finn et al., 2016). Our work is complementary to such approaches, building upon them to solve physical reasoning tasks.

**Model-based RL.** This class of RL approaches rely on building models of the environment of the agent to plan in. Such approaches typically use recurrent stochastic state transition models supervised with a reconstruction (Hafner et al., 2019; Ha & Schmidhuber, 2018; Hafner et al., 2020; Janner et al., 2019) or contrastive (Hafner et al., 2020) objective. Given the learned forward model, the planning is typically performed using a variant of cross entropy method (CEM) (Rubinstein, 1997; Chua et al., 2018). Our setup is similar to model-based RL, with the crucial difference being that we take only a single action, and the long-horizon dynamics we need to model is significantly more complex than typical RL control environments (Tassa et al., 2018; Todorov et al., 2012). Given the simplicity of our action space, we learn a value function over actions using the predicted rollouts, and use it to search for the optimal action at test time. Future work involving more complex or even continuous action spaces can perhaps benefit from learning a more sophisticated sampling approach using CEM.

## C    ROLLOUT ACCURACY IN CROSS-TEMPLATE SETTING

Similar to Figure 5 in the main paper, we show the forward-prediction accuracy on the *cross-template* setting in Figure 8. As expected, the accuracy is generally lower in the cross-template setting, showing that the models struggle to generalize beyond training templates. Otherwise, we see similar trends as seen in Figure 5.

It is interesting to note that `Dec` accuracy goes down and then up, similar to as observed in the within-template case. We find that it is likely because `Dec` is better able to predict the final position of the objects than the actual path the objects would take. Since it tends to smear out the object pixels when not confident of its position, the model ends up with lower accuracy during the middle part of the rollout.

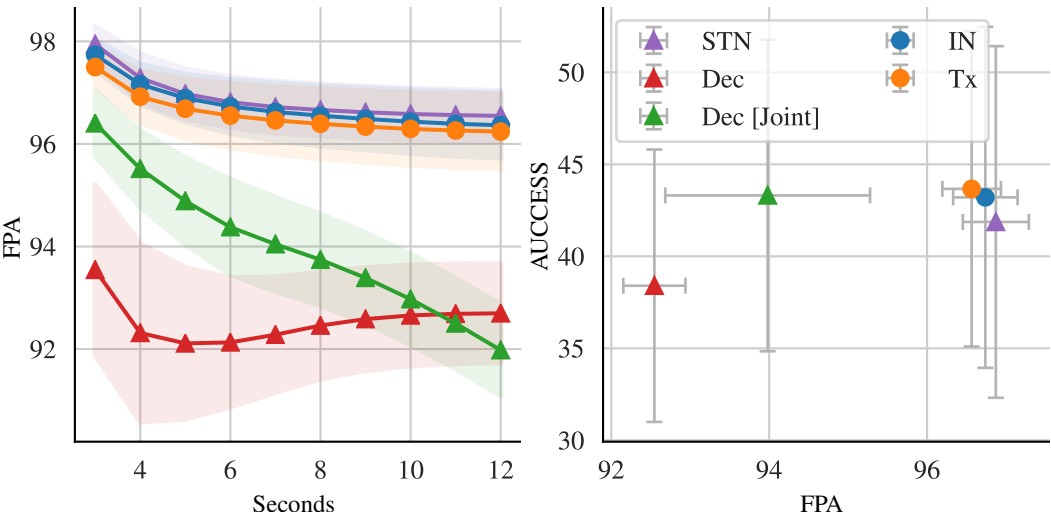

Figure 8: **Left:** Forward-prediction accuracy (FPA; $y$-axis) after $\tau'$ seconds ($x$-axis) rolling out five forward-prediction models. **Right:** Maximum AUCCESS value across roll-out ($y$-axis) as a function of forward-prediction accuracy averaged over $\tau' = 10$ seconds ($x$-axis) for five forward-prediction models. Shaded regions and error bars indicate standard deviation over 10 folds. Both shown for cross-template setting.

# D  OTHER TASK-SOLUTION MODELS ON LEARNED FORWARD-PREDICTION MODELS

## D.1  CONV3D-LATENT VS CONV3D-PIXEL, ON PIXEL-BASED FORWARD-PREDICTION MODELS

In Figure 9, we compare `Conv3D-Latent` and `Conv3D-Pixel` on learned pixel-based forward models. We find `Conv3D-Latent` generally performs better, especially for `Dec`, since that model does not produce accurate future predictions in terms of pixel accuracy (FPA). However, the latent space for that model still contains useful information, which the `Conv3D-Latent` is able to exploit successfully. Hence for pixel-based forward-prediction models, given the option between latent or pixel space task-solution classifiers, we choose `Conv3D-Latent` for experiments in the paper.

## D.2  TX-CLS VS CONV3D-PIXEL, ON OBJECT-BASED FORWARD-PREDICTION MODELS

In Figure 10, we compare the object-based task-solution model on learned object-based forward-prediction models. Similar to the observations with GT simulator in Figure 3 (main paper), object-based task-solution model (`Tx-Cls`) performed worse than its pixel-based counterpart (`Conv3D-Pixel`), even with learned forward-prediction models. Jointly training the object-based forward models with object-based task-solution model improves performance, as shown for the `IN` model, however it is still worse than using a pixel-based task-solution model on the object-based forward model. Hence, for experiments in the paper, we render the object-based models' predictions to pixels, and use a pixel-based task-solution model (`Conv3D-Pixel`). Note that the other pixel-based task-solution model, `Conv3D-Latent`, is not applicable here, as object-based forward-prediction models do not produce a spatial latent representation which `Conv3D-Latent` operates on. Moreover, training object-based models jointly with the pixel-based classifier is not possible in the absence of a differentiable renderer.

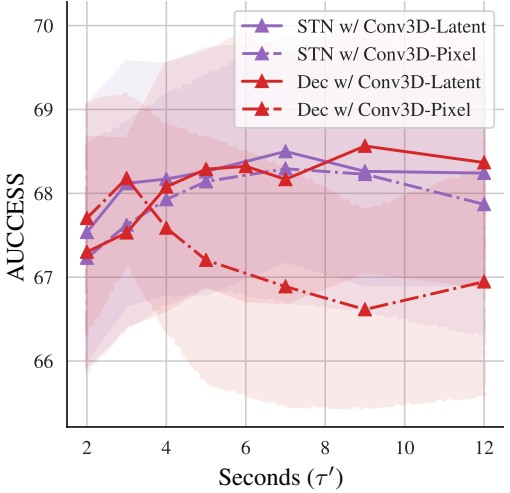 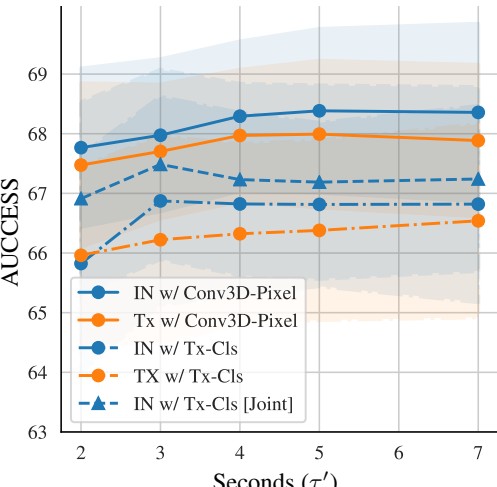

Figure 9: Comparison of `Conv3D-{Latent, Pixel}` classifiers on learned pixel-based forward-prediction models: `Dec` and `STN`, in the within-template setting. `Conv3D-Latent` performs as well or better than `Conv3D-Pixel`, and hence we use it for the experiments in the paper.

Figure 10: Comparison of `Tx-Cls` and `Conv3D-Pixel` classifiers on learned object-based forward-prediction models: `IN` and `Tx`, in the within-template setting. Since `Conv3D-Pixel` generally performed better, we use it for the experiments shown in the paper.

## E  FORWARD PREDICTION ACCURACY (FPA) METRIC

To compute FPA, we first zero out all pixels with colors corresponding to non-moving objects, in both GT and prediction. This ensures that any motion of non moving objects over other non moving objects or background will incur no reduction in the FPA score. Then, FPA for a frame of the rollout is defined as the percentage of pixels that match between GT and prediction. Hence, if any object of colors corresponding to moving objects (red, green, blue, gray) is at an incorrect position (either overlapping with static or non static objects/background), it would incur a reduction in FPA. A python-style code for the computation is as follows:

```python
def zero_out_non_moving_channels(img):
    is_red = np.isclose(img, RED)
    is_green = np.isclose(img, GREEN)
    is_blue = np.isclose(img, BLUE)
    is_gray = np.isclose(img, GRAY)
    is_l_red = np.isclose(img, L_RED)
    img[~(is_red | is_green | is_blue | is_gray | is_l_red)] = 0.0
    return img

def fpa(prediction, gt):
    """
    prediction: predicted frame of dimensions (H, W, 3)
    gt: Corresponding GT frame of dimensions (H, W, 3)
    """
    prediction = zero_out_non_moving_channels(prediction)
    gt = zero_out_non_moving_channels(gt)
    is_close_per_channel = np.isclose(prediction, gt)
    all_channels_close = is_close_per_channel.sum(
        axis=-1) == prediction.shape[-1]
    frame_size = gt.shape[0] * gt.shape[1]
    return np.sum(all_channels_close) / frame_size
```

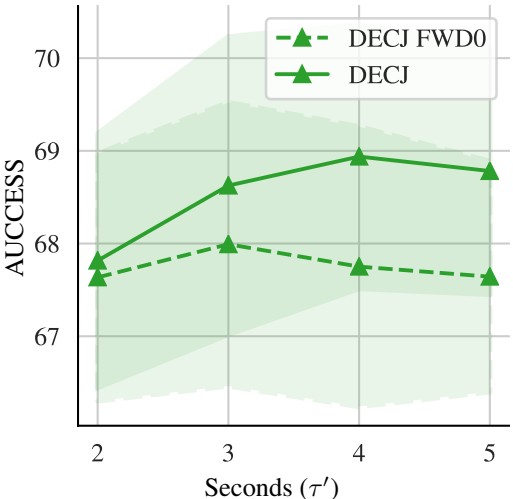
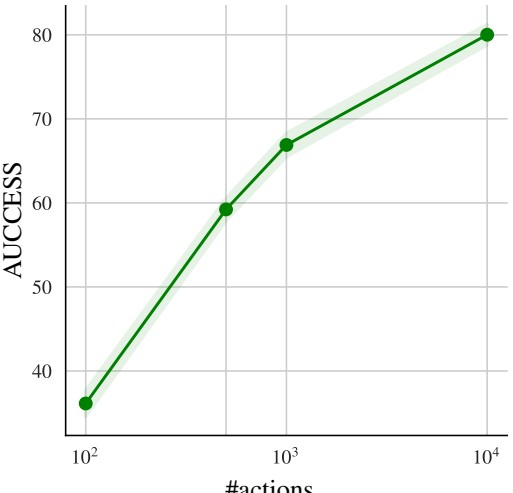

Figure 11: **Effect of setting forward prediction loss to 0 in `Dec [Joint]`.** The performance is stagnant with the rollout if loss on the future prediction is set to 0, as expected. The model performs comparably to a model without forward prediction.

Figure 12: **Performance of `Dec [Joint] 1f` at different number of actions.** The performance varies nearly linearly with number of actions ranked.

## F  JOINT MODEL WITH ONLY TASK-SOLUTION LOSS

For our best joint model, `Dec [Joint]`, we evaluate the effect of setting the forward prediction loss to 0. As seen in Figure 11, the model performs about the same as it would without a forward model, obtaining similar performance at different number of rollout seconds ($\tau'$). This further strengthens the claim that forward prediction leads to the improvements in performance (as also evident from the increase in AUCCESS on increasing $\tau'$), as opposed to any changes in parameters or training dynamics.

## G  PERFORMANCE WITH DIFFERENT NUMBER OF ACTIONS RANKED

Similar to Figure 4 in Bakhtin et al. (2019), we analyze the performance of our best model, `Dec [Joint] 1f`, at different number of actions being re-ranked at test time, in Figure 12. We find the performance varies nearly linearly with number of actions upto 10K actions, similar to the observations in Bakhtin et al. (2019).

## H  TEMPLATES RANKED BY FPA

Figure 13 shows the easiest and hardest templates for each forward model. We find some of the hardest ones are indeed the ones that humans would also find hard, such as the template involving a see-saw system or complex extended objects like cups.

## I  HYPERPARAMETERS

All experiments were performed using upto 8 V100 32GB GPUs. Depending on the number of steps the model was rolled out for during training, the actual GPU requirements were adjusted. The training time for all forward-prediction models was around 2 days, and the task solution models took upto 4 days (depending on how far the forward-prediction model was rolled out). Our code will be made available to reproduce our results.

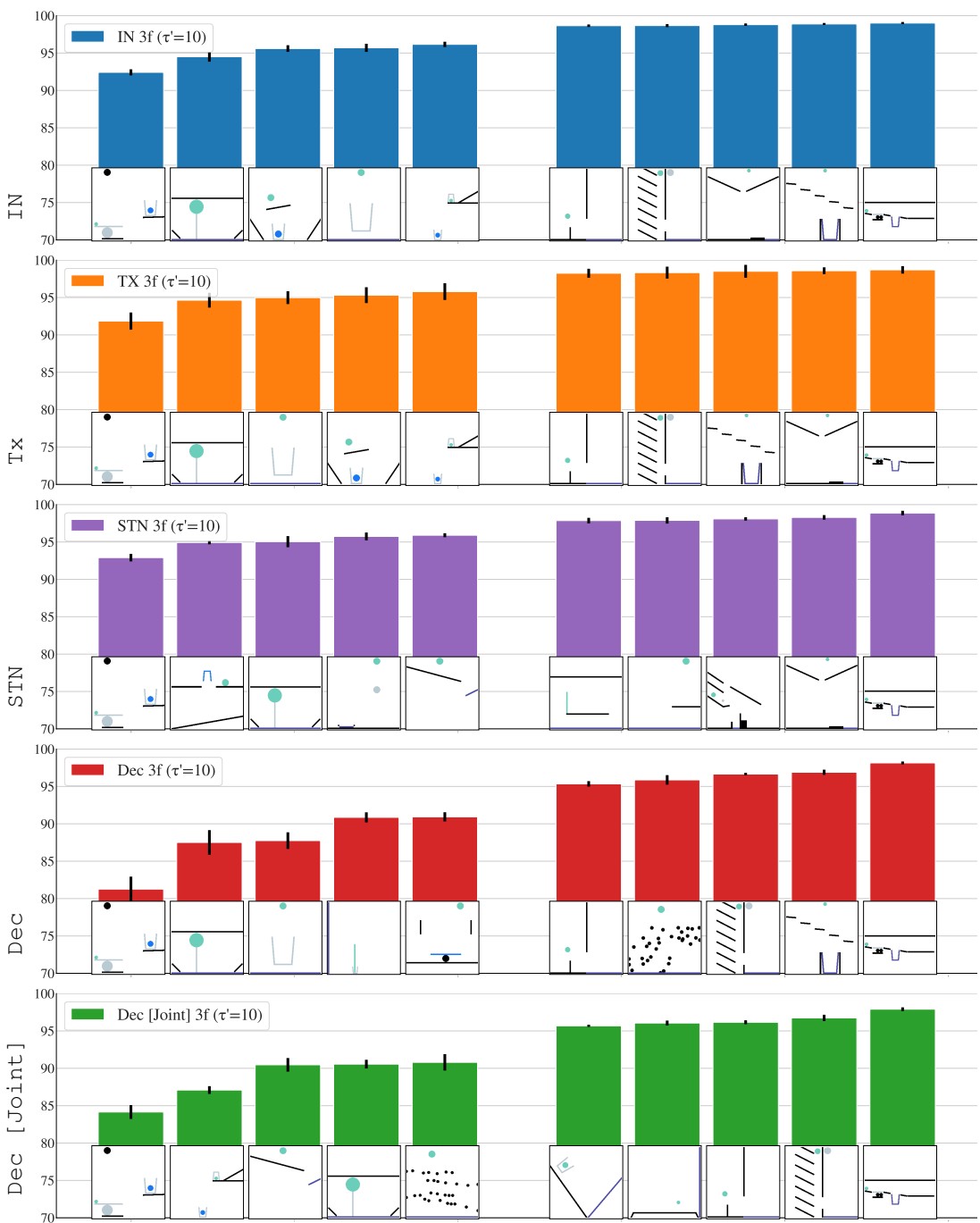

Figure 13: **Easiest and hardest templates in terms of FPA.**

## I.1 FORWARD-PREDICTION MODELS

We train all object-based models with teacher forcing. We use a batch size of 8 per GPU over 8 gpus. For each batch element, we sample clips of length 4 from the rollout, using the first three as context frames and the 4th as the ground truth prediction frame. We train for 200K iterations. We add Gaussian noise sampled from a $\mathcal{N}(0, 0.014435)$ distribution to the training data, similar to (Battaglia et al., 2016). We add noise to 20% of the data for the first 2.5% of training, decreasing the percentage of data that is noisy to 0% over the next 10% of training. The object based forward models only make

predictions for dynamic objects, and use a hard tanh to clip the predicted state values between 0-1. The models don't use the state of static objects or the angle of ball objects when calculating the loss. For angles, we compute the mean squared error between the cosine of the predicted and ground truth angle. We now describe the other hyperparameters specific to each object-based and pixel-based forward-prediction model.

- **IN (object-based):** We train these models using Adam and a learning rate of 0.001. We use two interaction nets, one that makes predictions based on the last two context frames, the other that makes predictions based on the first two context frames. Using the same architecture as (Battaglia et al., 2016), the relation encoder is a five layer MLP, with hidden size 100 and ReLU activation, that embeds the relation into a dimension 50 vector. The aggregated and external effects are passed through a three layer MLP with hidden size 150 and ReLU activation. Each interaction net makes a velocity prediction for the object, and the results are concatenated with the object's previous state and passed to a three layer MLP with hidden size 64 and ReLU activation to make the final velocity prediction per object. The predicted state is a sum of the velocity and the object's previous state.

- **Tx (object-based):** We train these models using Adam and a learning rate of 0.0001. We use a two layer MLP with a hidden size of 128 and ReLU activation to embed the objects into a 128 dimensional vector. A sinusoidal temporal position encoding (Vaswani et al., 2017) of time $t$ is added to the features of each object. The result is passed to a Transformer encoder with 8 heads and 6 layers. The embeddings corresponding to the last time step are passed to the final three layer MLP with ReLU activations and hidden size of 100 to make the final prediction. The model predicts the velocity of the object, which is summed with the object's last state to get a new state prediction.

- **STN (pixel-based):** We also train these models with teacher forcing. Since pixel-based models involve a ResNet-18 image encoder, we use a batch size of 2 per GPU, over 8 GPUs. For each batch element, we sample clips of length 16 from the rollout, and construct all possible sets with 3 context frames and the 4th ground truth prediction frame. The models were trained using a learning rate of 0.00005, adam optimizer, with cosine annealing over 100K iterations. The scene was split into objects using the connected components algorithm, and we split each color channel into upto 2 objects. The model then predicts rotation and transformation for each object channel, which are used to construct an affine transformation matrix. The last context frame is transformed using this affine matrix to generate the predicted frame, which is passed through the image encoder to get the latent representation for the predicted frame (i.e. for STN, the future frame is predicted before the future latent representation).

- **Dec (pixel-based):** For these models, we do not use teacher forcing, and use the last predicted states to predict future states at training time. We use a batch size of 2 per gpu over 8 gpus. For each batch element, we sample a 20-length clip from the simulator rollout, and train the model to predict upto 10 steps into the future (note with teacher forcing, models are trained only to predict 1 step into the future given 3 GT states). The model is trained for 50K iterations with a learning rate of 0.01 and SGD+Momentum optimizer.

- **Dec [Joint] (pixel-based):** For this model, we train both forward-prediction and task-solution models jointly, with equally weighted losses. For this, we sample 13-length rollout, always starting from frame 0. Instead of considering all possible starting points from the 13 states (as in Dec and STN), we only use the first 3 states to bootstrap, and roll it out for upto 10 steps into the future. We only incur forward-prediction losses for upto the first 5 of those 10 steps, we observed instability in training on predicting for all steps. Here we use a batch size of 8/gpu, over 8 gpus. The model is trained with learning rate of 0.0125 with SGD+Momentum for 150K iterations. The task-solution model used is Conv3D-Latent, which operates on the latent representation being learned by the forward-prediction model.

## I.2 Task-solution models

For all these models, we always sample $\tau = 3$ frames from the start point of each simulation, and roll it out for different number of $\tau'$ states autoregressively, before passing through one of these task-solution models.

- **Tx-Cls (for object-based):** We train a Transformer encoder model on the object states predicted by object-based forward-prediction models. The object states are first encoded using a two layer MLP with ReLU activation into an embedding size of 128. A sinusoidal temporal position

encoding (Vaswani et al., 2017) of time $t$ is added to the features of each object. The result is passed to a Transformer encoder which has 16 heads and 8 layers and uses layer normalization. The encoding is passed to three layer MLP with hidden size 128 and ReLU activations to classify the embedding as solved or not solved. We use a batch size of 128 and train for 150K iterations with SGD optimizer, a learning rate of 0.002, and momentum of 0.9.

- **`Conv3D-Latent` (for pixel-based):** We train a 5-layer 3D convolutional model (with ReLU in between) on the latent space learned by forward-prediction models. We use a batch size of 64 and train for 100K iterations with SGD optimizer and learning rate of 0.0125.

- **`Conv3D-Pixel` (for both object and pixel-based):** We train a 2D + 3D convolutional model on future states rendered as pixels. We use a batch size of 64 and train for 100K iterations with SGD optimizer, learning rate of 0.0125, and momentum of 0.9. This model consists of 4 ResNet-18 blocks to encode the frames, followed by 5 3D convolutional layers over the frames' latent representation, as used in `Conv3D-Latent`. When object-based models are trained with this task-solution model, we run the forward-prediction model and the renderer in the data loader threads (on CPU), and feed the predicted frames into the task-solution model (training on GPU). We found this approach to be more computationally efficient than running both forward-prediction and task-solution models on GPU, and in between the two, swapping out the data from GPU to CPU and back, to perform the rendering on CPU.

