# OpenReview forum: "Forward Prediction for Physical Reasoning"
_ICLR.cc/2021/Conference — Reject_

### Official Review · AnonReviewer1 · 2020-10-28
**A quantitative evaluation with unclear takeaways**

**Rating:** 5
**Confidence:** 4

**Review:**

== Update ==

After reading the rebuttal I have left my score unchanged. I appreciate the clarifications, but am very concerned about the result that the pixel-based models perform worse than the identitiy function in the FPA metric. When a model fails a sanity check like that, I believe the causes and consequences need to be thoroughly investigated. In its current form, the paper does not provide that.

== Original Review ==

The paper evaluates different methods for forward prediction on the PHYRE benchmark for physical
reasoning. Using either ground-truth simulator state or pixel-based observations as input, both
object-based and pixel-based methods are trained to predict future observations. By combining the
physics models with a classifier predicting task success, search agents for the PHYRE benchmarks are
constructed. The results show that the success rate of agents based on the forward prediction models
is slightly higher than that of an agent without forward prediction, but still far lower than that
of one with access to the oracle simulator. Incorporating forward prediction was found to be most
helpful on a subset of tasks with comparatively high complexity and object count, but was of little
help in simpler tasks. Overall, the reported improvement over a previously proposed DQN agent is
narrow, but not significant. The pixelwise accuracy of forward prediction was not found to be
correlated with task performance of the resulting agent.

Strenghts:
 * The paper is one of the first to offer quantitivate results on the PHYRE benchmark, and the first
   (to my knowledge) to do so for forward prediction methods.
 * The selection of methods is sensible and their way they were applied to the benchmark is clearly
   described.
 * The paper is well written, and largely easy to follow.

Weaknesses:
 * The paper does not introduce novel methods or techniques, but evaluates existing ones.  As such,
   it offers little technical novelty.
 * Qualitativly, based on the visualizations referenced in Appendix A, none of the examined methods
   generate convincing, physically plausible rollouts. All exhibit glitches, such as objects
   overlapping, passing through each other, or changing in size. The joint deconvolutional model in
   particular, which performed best quantitatively, generates severly distorted shapes.
   These shortcomings are not discussed or analyzed in depth, even though they potentially explain
   much of the empirical results.
 * It is not clear to me why joint training is described as a unique capability of the
   deconvolutional model, e.g. by stating that it is its "key advantage". There is no conceptual
   reason why joint training should not be possible for the other models, and Figure 10 in the
   appendix seems to indicate that it is even helpful. Given the spurious rollouts mentioned above,
   it seems plausible that the performance gain of the joint model may have largely come from its
   greater model capacity, which may be utilized to improve the accuracy of the task-solution model
   while bypassing forward prediction.

While I think that not every paper needs to introduce new techniques, and that papers providing
analysis or negative results can be valuable contributions, in this case, I find it difficult to
derive actionable insights from the presented study. It shows that physics prediction on an
environment like PHYRE is still difficult, but offers few pointers as to what is going wrong or how
it might be improved. As a results, I view it as slightly below the threshold in its current state.

Questions:
 * How does the forward-prediction accuracy (FPA) (Fig. 5) compare to naive baselines, e.g. the identity
   function? Given that not all scenes even exhibit movement, it is hard to judge how strong the
   reported values really are.
 * It is stated that FPA is only computed for the pixels corresponding to dynamic objects. Does this
   refer to dynamic pixels in the ground-truth, the model's prediction, or both?
 * How well does the architecture of the joint model perform when it is purely trained using the
   task-solution loss, without the prediction loss? If it is no better than the no-forward model,
   then that would disprove my concern regarding model capacity mentioned above.

---

> ### Author Response · Authors · 2020-11-24
> **Author response**
>
> We thank the reviewer for their time and insightful feedback. We are glad they found our paper well written, providing the first forward prediction results on the PHYRE benchmark. We have incorporated the suggestions in the updated paper, and address specific questions next.
>
> ## Response to Questions
>
> ### FPA comparison to naive baselines, like identity function
>
> That is a great suggestion, and we have updated the paper Fig 5 (left) in the paper with an identity baseline. We find the identity baseline does worse than our object-based models, but better than the Dec models. This is expected from the rollout visualizations, where we can see that Dec models do not ensure conservation of mass. Note that while there might be many static objects in a scene, there are at least two moving objects in each scene: the red ball that is added, and the green object that needs to be made to touch the blue object. Since the FPA metric only considers moving objects, it is not affected by the static objects in the scene.
>
> ### Computation of FPA
>
> Apologies for the confusion, we have provided detailed description and pseudo code for FPA computation in Appendix E. It refers to dynamic object pixels in both. Specifically, we zero out all non-dynamic color pixels, and then compute the percentage of pixels that match in GT and prediction. Hence, if a black object (static) is moved by the model to occupy a different patch of white background (static), it will not incur any reduction in FPA. However, any of the colors corresponding to moving objects (red, green, blue, gray) is at an incorrect position (either overlapping with static or non static objects), it would incur a reduction in FPA.
>
> ### Performance of joint model with only task-solution loss
>
> This is a great suggestion, and we have added this result to Appendix F. We show the performance of the model trained with 0 weight on the future prediction loss, for $\tau’=2,3,4$. The model without the future prediction loss performs the same irrespective of $\tau’$ (i.e. the length of rollout that the task-solution model sees), since the rollouts are not informative as they are not being learned. Hence, it is no better than a no-forward model. This proves that the gains are truly due to the forward prediction, and not due to additional model capacity or changed training dynamics.
>
> Please note that the capacity for all our models is the same for all values of $\tau’$; no new parameters are added and the model is rolled out autoregressively. Hence, the improvement we see on going from $\tau’=2$ to $\tau’=4$ already shows that the forward prediction leads to the gains, and it is not from any additional parameters.
>
> ## Response to other concerns raised
>
> ### Technical novelty and actionable insights
>
> That is a valid concern. Please refer to our common response to all reviewers for a discussion on this point.
>
> ### Visualizations, glitches in rollouts
>
> That is a valid concern. To better understand the source of glitchy rollouts, we have now added visualizations of one of our models with high FPA score, STN, when trained and tested on a __single template__ [here](http://fwd-pred-phyre.s3-website.us-east-2.amazonaws.com/within-temp18only/) (and also to Appendix A). The rollouts here look a lot more accurate and physically plausible.
>
> This further strengthens our claim that the primary reason our rollouts look glitchy is because forward modeling on PHYRE is a significantly harder problem: a single model needs to capture all the 25 different templates; and in the cross-template setting, needs to generate rollouts of physical environments never seen before. Note that even the most recent work, including latest particle-based methods (Sanchez-Gonzalez et al. (2020)), do not test for this level of generalization. Most existing papers use small variations of training templates for testing, which is why we believe our analysis on PHYRE is significant and moves the community towards more rigorous testing of physical reasoning and forward prediction. Hence, the lower fidelity rollouts when training on all templates, especially in cross template testing setting, is expected.
>
> ### Joint training of object-based models
>
> Indeed, we are able to train object-based models jointly, as shown in Figure 10. We have reworded the “key advantage” sentence to avoid any confusion. However, given the stronger performance of pixel-based task-solution models, which are only applicable to pixel-based methods for joint training, does confer an advantage to pixel-based methods leading to the overall best performance. As our earlier result with 0 weight on the prediction loss has shown, the gains do not come from the additional model capacity, but from the forward prediction. Even though the predictions may look glitchy visually, the prediction loss provides enough signal to learn a **better latent representation**, which is used by the task-solution model to improve the end task performance.

---

### Official Review · AnonReviewer3 · 2020-10-29

**Rating:** 5
**Confidence:** 3

**Review:**

Rebuttal Update #####
I thanks the authors for their responses to my questions. They were very helpful, and I think the work, when explored further, would be a great submission to a future conference. However I do share sentiments with other reviewers about following set of issues.

(1) The novelty is a bit limited as the paper did not introduce any novel technique approaches. While not every paper needs to propose a new method, a more in-depth analysis of the benchmarked approaches may be needed to provide insights into how existing methods fail and how we can improve them.

(2) The scope of this paper is a bit narrow where the authors only evaluated the methods on PHYRE that is fully-observable and only contains open-loop tasks with rigid objects of simple shapes in 2D space.


######
Strengths:
The paper is pretty well written.
I enjoyed the analysis in the paper and thought the experiments were relatively thorough.

Weaknesses
The overall approach seems to be rather incremental,  with many past papers on MPC control on some type of learned dynamics model, some with reward functions and others with value functions. For example see [1]. Dynamics learning also uses standard architectural components.

Why is it that better pixel prediction accuracy in object models actually lead to lower AUCCESS? Is it because it is harder to learn the goal model?

Could authors provide more intuition on the templates in which an agent does poorly? Is it similar to the tasks that humans typically perform poorly?

It may also be more interesting to learn more stochastic dynamics model, and average rollouts over a large number of samples from the dynamics model.

The performance gains over a DQN agent appear to be relatively minor

Along this line, it might be more interesting to explore this method in combinations with some type of value function over future states.

I think it would beneficial for the community if source code for the submission was provided, as it still seems there are many free parameters that seem difficult to describe in the paper.

Minor:
[2] might a somewhat related reference that might be good to add.

[1] Chelsea Finn et. al. Deep Visual Foresight for Planning Robot Motion
[2] Yilun Du, Karthik Narasimhan. Task-Agnostic Dynamics Priors for Deep Reinforcement Learning. ICML 2019
might also be worth citing about learning physical-reasoning.

---

> ### Author Response · Authors · 2020-11-24
> **Author response**
>
> We thank the reviewer for their time and insightful feedback. We are glad they found our paper well written, enjoyed our analysis and found our experimentation thorough. We have incorporated the suggestions in the updated paper, and address specific questions next.
>
> ### Technical novelty
>
> That is a valid concern. Please refer to our common response to all reviewers for a discussion on this point.
>
> ### Examples of templates where the agent does poorly
>
> Great suggestion! We have added this analysis to Appendix H, for every forward model in our paper. Indeed, we find some of the hardest templates are the ones that humans would also find hard, such as the templates involving a see-saw system balanced on a ball, or those involving complex extended objects like cups.
>
> ### Why does better pixel accuracy in object models lead to lower AUCCESS? Is it because it is harder to learn the goal model?
>
> That is indeed correct. The task-solution model in object-space tended to perform worse than its pixel-based counterparts. This is not entirely surprising in the PHYRE environment, where evaluating whether two objects are touching can be quite hard in object-space. For instance, to evaluate contact between complex extended objects like a cup and a rod, the task-solution model will have to solve a complex function to infer whether any pixel of one object touches any pixel of the other. This, instead, is much easier in the pixel space, and a convnet can easily check if any pixels on the two objects are touching.
>
> ### Stochastic dynamics model
>
> That is a great suggestion. Following most recent work in physical reasoning, we limited our exploration in this work to deterministic methods. We did try some initial experiments for object based methods, making them stochastic by discretizing the output space and predicting a softmax distribution over the possible future states, as opposed to regressing. However, we did not see promising gains. Nevertheless, we believe this is an interesting direction for future work.
>
> ### Source Code
>
> We are planning a full code release together with the publication of the paper. We were unable to share in the submission stage due to the large number of experiments that would need careful anonymization.
>
> ### Other
>
> Thank you for pointing out the reference, we have added it.

---

### Official Review · AnonReviewer2 · 2020-11-05
**Thorough evaluation, but the novelty is a bit limited, and the observations may be very specific to the testing environments.**

**Rating:** 5
**Confidence:** 4

**Review:**

=== Summary

This paper investigates the performance of several state-of-the-art forward-prediction models in the complex physical-reasoning tasks of the PHYRE benchmark. The authors have provided thorough evaluations of the models by ablating on different ways of representing the state (object-based or pixel-based), forms of model class (Interaction network, Transformers, Spatial transformer networks, etc.), and specific evaluations settings (within-template or cross-template), from which they made several interesting observations. For example, forward predictors with better pixel accuracy do not necessarily lead to better physical-reasoning performance. Their best-performing model also sets a new state-of-the-art on the PHYRE benchmark.


=== Strengths

This paper targets a very challenging task, the PHYRE benchmark, where the simulation results can be very sensitive to small changes in the action (or the world), and their best-performing model from this paper outperforms the prior state-of-the-art methods on this benchmark.

This paper provides a thorough evaluation of the forward-prediction models by considering different state representation and model class, resulting in some interesting observations about the relationship between the modeling choices and the physical-reasoning performance.

The videos in the supplemental materials are very illustrative and provide good qualitative comparisons between the methods.


=== Weaknesses

My primary concern of this paper is the novelty is a bit limited. This paper does not propose any new method, but mainly focus on comparing several existing forward-prediction approaches by assessing their ability to perform physical reasoning on the PHYRE benchmark. Although the best model achieves a new state-of-the-art, I would not consider it to be particularly novel.

The proposed method seems to be very specific to the PHYRE benchmark, which, although challenging but only contains open-loop tasks with rigid objects of simple shapes in 2d space. It is hard to know whether this paper's observations are still valid in more diversified and complicated environments. For example, this paper suggests that "pixel-based models are more helpful in physical reasoning" than object-based models, which may not be true if we apply the methods in three-dimensional environments where pixel-based models could suffer from occlusions and a poorer estimation of the 3d location and geometry of an object.

Even if the forward model is reasonably accurate, the control problem can still be very challenging. Given that the results are very sensitive to small variations in the initialization, the task-solution model and the search strategy proposed in this paper may require extensive samples to find a suitable solution to circle around bad local minimums. What is the distribution of the sampling space? How does the performance change with respect to the number of samples? How long does the process take?

When training the forward-prediction model, using a fully-connected graph to model the contacting events between the objects can sometimes lead to unsatisfying results. This is because neural networks are essentially a continuous function, whereas contacts are inherently discontinuous. As shown in the videos accompanying this paper, IN sometimes misses or smoothes the contacts, which is also the reason why some other works choose to add edges between constituent components only when they are close enough [1,2]. The object-based model may have a better performance if the graph is built dynamically.

When constructing the training set, the authors "sample task-action pairs in a balanced way: half of the samples solve the task and the other half do not." How did the authors specify the sampling space? How likely will a random sample solve the task, given the sensitivity of the simulation results to the input variations?

In Figure 5, are there any intuitive explanations of why the red curve first decreases and then increases?


[1] A Compositional Object-Based Approach to Learning Physical Dynamics, ICLR 2017

[2] Learning to simulate complex physics with graph networks, ICML 2020


=== Post rebuttal

Having read the rebuttal and the reviews from other reviewers, my rating remains the same (5: Marginally below acceptance threshold). Many of the reviewers share similar concerns:

(1) The novelty is a bit limited as the paper did not introduce any novel technique approaches. While not every paper needs to propose a new method, a more in-depth analysis of the benchmarked approaches may be needed to provide insights into how existing methods fail and how we can improve them.

(2) The scope of this paper is a bit narrow where the authors only evaluated the methods on PHYRE that is fully-observable and only contains open-loop tasks with rigid objects of simple shapes in 2D space.

(3) The conclusion that the pixel-based model does better than the object-based counterparts may be a bit controversial. The observation is very specific to the methods and environments the authors were using and may not hold when generalizing to more complicated partially-observable or 3D scenarios.

---

> ### Author Response · Authors · 2020-11-24
> **Author response**
>
> We thank the reviewer for their time and insightful feedback. We are glad they find our work addresses a very challenging task with thorough evaluation resulting in interesting observations. We have incorporated their suggestions in the updated paper, and address specific questions next.
>
> ### Technical novelty
>
> That is a valid concern. Please refer to our common response to all reviewers for a discussion on this point.
>
> ### Generalizing to 3D environments
>
> That is a great point, and we agree that our analysis is specific to PHYRE, as we wanted to keep this work focused on the challenging setup posed in these physical puzzle benchmarks. To our knowledge, there does not exist any 3D environment testing for a similar level of generalization and implicit, task-based physical reasoning; as both Tools (Allen et al. 2020) and PHYRE, only operate on a 2D environment. Extension of our analysis to more complex 3D environments is definitely an important direction for future work. We have added a note in our discussion.
>
> ### Analysis of the control problem, sampling space
>
> That is a great point, and we have now added the performance of our best (Dec [Joint] 1f) model, at different numbers of sampled actions, to Appendix G. We use uniform random sampling to sample actions at test time, in order to be consistent and comparable to the baselines from the original paper (Bakhtin et al. (2019)), and the number of actions ranked in benchmark setting is also consistent with prior work. Regarding runtime, on average it took 1.6s to rank each action, and the performance varied linearly with the number of actions. We note that model inference was mostly data bound and can be trivially parallelized and sped up through further optimizations of the data loader. Our method can be further improved with learned sampling approaches such as cross-entropy method, and would be interesting to explore in future work.
>
> ### Dynamic edges between objects
>
> That is a great suggestion. For the purpose of this work, we wanted to focus our attention on the most standard paradigms for forward prediction. Indeed, dynamically weighting interactions could help further improve performance, and we would love to explore in future work.
>
> ### Sampling of tasks in a balanced way
>
> Balanced sampling is done only during training to ensure the model sees an equal number of action samples that solve the task and those that don’t. We do so similar to the prior work (Bakhtin et al. (2019)), where a cached set of (action, output) is used for training, and half the batch is sampled from actions where output=solved, and the other half where output=not solved.
>
> ### How likely would a random action solve the task?
>
> We report that number in Table 1 (RAND agent, as defined in Bakhtin et al. (2019)).
>
>
> ### Fig 5, explanation of why the red curve decreases and then increases
>
> Great point. We have an explanation on this in Appendix C. We think it is likely because Dec is better able to predict the final position of the objects than the actual path the objects would take. Since it tends to smear out the object pixels when not confident of its position, the model ends up with lower accuracy during the middle part of the rollout.

---

### Official Review · AnonReviewer4 · 2020-11-05
**Interesting work with surprising conclusion and supported by convincing experiments.**

**Rating:** 6
**Confidence:** 5

**Review:**

Summary: This work investigates how a forward prediction model helps the physical reasoning task. The authors propose two variants of forward prediction model, i.e., object-based and pixel-based. The authors also design a classification model, taking predicted results as inputs, to evaluate the efficiency of the prediction model. Task-related metric, i.e., FPA, is proposed to assess the performance. An interesting conclusion, indicated by the authors, is that an accurate predictor does not necessary help the success of physical reasoning.

Pros:

+ Quality: This paper is well written. The authors introduce the motivation behind this work, i.e., to investigate the usefulness of the forward prediction model in the physical reasoning task. The method part is clearly presented with sufficient details, including the model architecture and input/output description. The experiment part is well organized in the format of corresponding concerns related to this work.

+ Clarity: I have carefully checked the manuscript and supplementary material. The whole paper is overall easy to understand. It is enjoyable to read this work. The majority of the details needed to be clearly presented are considered, e.g., the experimental settings, illustration of the newly introduced metric, the description of the figure and table.

+ Significance: The conclusions claimed by the authors are all supported by convincing evidence. The performance boosting on physical reasoning of complex scene proves the usefulness of the proposed method. Meanwhile, authors also point out that the generalization issue, i.e., generalizing to other template, is still challenging. The authors also show that the necessity of developing an accurate prediction model is still open to discussion, which is supported by convincing evidence.

Cons:

- Originality: My major concern lies in the novelty. In my point of view, this work is more like an analysis project. This most significant part is the design of the full pipeline connecting the prediction model and the downstream task, i.e., physical reasoning. The detailed architecture of prediction model and training scheme generally follows the basic configuration of prediction task. However, considering that the main focus of this paper is not pursuing a better performance of prediction model, I think this part is not so important.

---

> ### Author Response · Authors · 2020-11-24
> **Author response**
>
> We thank the reviewer for their time and encouraging review. We are glad they found our work well written and enjoyable to read, with convincing evidence for the reported claims. We agree with their assessment on originality, our main focus is not to propose a new model, but a rigorous analysis of standard approaches when applied to a challenging benchmark like PHYRE. We have added a discussion on that point in the common response to all reviewers.

---

### Official Review · AnonReviewer5 · 2020-11-10
**Some findings are interesting but the scope of the paper is very narrow.**

**Rating:** 5
**Confidence:** 3

**Review:**

This paper discusses the importance of forward prediction in physical reasoning, and particularly in the PHYRE benchmark: a dataset of physical tasks where the agent is asked to place a ball of a chosen radius in a 2d environment. The authors build a classifier that, given an initial state, predicts the probability of success in a given task. `Given the classifier, one can solve PHYRE tasks by sampling actions uniformly at random, and using actions that achieve good scores under the classifier. The classifier consists of an encoder, dynamics model, decoder, and a final classifier. The authors then evaluate two types of dynamics models (interaction nets and transformers) that act on two different types of representations (either images or object states).
The paper concludes that forward prediction is, in fact, beneficial to solving PHYRE tasks, with the best model being a conv-net operating in the pixel space achieving the a SOTA on PHYRE.

The introduced method delivers a new SOTA and some of the reported results are interesting. The evaluation is done on only one dataset, but it is nonetheless quite extensive. I find the paper interesting, but I have mixed feelings about it for I find some of its aspects deeply unsatisfying:
1. The paper considers only a single fully-observable 2D dataset and many findings cannot be generalized beyond this domain.
It is therefore unclear if the paper's conclusion would transfer to other domains, partially-observable settings, or even datasets. Specifically, I am concerned about the fact that in this paper's experiments the pixel-based model does better than the object-based one. This is surprising, as authors correctly note, but I am concerned about the justification: "it is easier to determine whether a task is solved in a pixel-based representation than in an object-based one" as stated in the 2nd paragraph of the intro. This would not be the case in any partially-observed environment, or even fully-observed 3D where some parts of the objects are not visible due to self-occlusion.

2.  A large body of related is omitted from the paper and not even mentioned.
When the authors write say "models that operate on object representation" in the abstract, I immediately think about models that infer object representations end-to-end like AIR (Eslami et. al.), MONET (Burgess et. al.), GENESIS (Engelcke et. al) and similar but equipped with transition models (e.g. SQAIR (https://arxiv.org/abs/1806.01794), SILOT (https://arxiv.org/abs/1911.09033) or RNEM (https://arxiv.org/abs/1802.10353)) that seem to fit the paper's setting ideally. However, these models are not considered in this work, and are not even mentioned in the paper. I consider this a severe oversight. This is made even worse by the fact that the best model of the paper is the one trained end-to-end, which is impossible with the considered object-based models, but would be possible with the models I mention above. Please at least discuss these models in the related works and justify why you decided to not use them. Ideally, at least some of these models would be included as baselines.

3.  The paper lacks clarity in places and some of the modelling choices are non-obvious and lack justification.
The best example of the lack of clarity is the following. All models considered in the paper supposedly can be described in terms of encoders, dynamics models, state decoders and task-solution models as shown in Figure 2. The paper, however, fails to explicitly mention what encoders and decoders are used for the considered dynamics models.
As for the modelling choices, a simple example is that of "Objects in PHYRE can have seven different colors; hence, the input of the network
consists of seven channels" in Sec 4.1. Why is the image represented like that? Is that something that comes with the dataset or was it done in a preprocessing step? It is not natural to represent images this way, though I understand why this is the case here.
Another example is that of inputs to dynamics models and the task-solution models: they all seem to have access to the whole history of inputs and/or latent states. If the task-solution model sees all the data, why does it even need a dynamics model?

4. Solving PHYRE tasks is done by sampling actions uniformly at random and evaluating their success probability under the model, and the authors choose to try K=1000 such actions. While not wrong in principle, it is conceptually unsatisfying that such a high number of random actions is used. Why not train a policy that would learn to sample appropriate actions for a given task?

 As it stands, I think that this paper requires a little bit more work before it can be accepted. I would be happy to change my score if these issues are addressed.

Other remarks:
- in the 2nd sentence of the intro there is an object change from "humans" to "we" that is a bit confusing.
- in the abstract you also say that "[...] these improvements are contingent on the training tasks being similar to the test tasks", which is fair, but is hardly surprising and is an issue with deep learning at large: models generally fail to generalize out of distribution. If so, why report in in the abstract?
- also in the abstract you say that "Surprisingly, we observe that forward predictors with better pixel accuracy do not necessarily lead to better physical-reasoning performance". https://arxiv.org/abs/1802.03006 makes a similar observation in the context of RL, and it might be a good idea to cite it. I think this finding was reported in a number of papers now and is not "surprising" anymore.

---

> ### Author Response · Authors · 2020-11-24
> **Author response**
>
> We thank the reviewer for their time and insightful feedback. We are glad they liked our extensive experimentation and found our paper and results interesting. We have incorporated the suggestions in the updated paper, and address the specific questions next.
>
> ### Generalizing conclusions to partially observed or 3D environments
>
> That’s a valid point, and we agree that our analysis is specific to PHYRE, as we wanted to keep this work focused on the challenging setup posed in the physical puzzle benchmarks. We have updated the paper introduction text to specify that that observation is specific to fully observable 2D environments like PHYRE. To our knowledge, there does not exist any 3D environment testing for a similar level of generalization and implicit, task-based physical reasoning; as both Tools (Allen et al. 2020) and PHYRE, only operate on a 2D environment. Extension of our analysis to more complex 3D or partially observable environments is definitely an important direction for future work. We have also added a note in our discussion.
>
>
> ### Prior work on object discovery
>
> Thank you for pointing out these papers, we have added a discussion in the related work. These are definitely interesting and relevant, however in the 2D PHYRE environment, the object discovery process is actually quite simple. For instance, our STN model is able to break the scene into objects using a simple connected components algorithm. Approaches like MONET are designed to decompose more complex scenes, often in 3D, which we believe may not be needed for an environment like PHYRE. Moreover, our methods that operate on object-based representation directly have access to the ground truth object state, and do not need to infer it from the scene. Hence, our focus is on learning the dynamics, which we believe is the core challenge in PHYRE.
>
> Regarding end-to-end training of object-based models, we do provide an end-to-end version of object-based models in Appendix D.2, using an object-based task-solution model. While joint training does improve over separately trained object-based models, it still does not outperform pixel-based methods, since the object-based task-solution model lacks in performance compared to pixel-based one.
>
> That being said, these models are interesting and worth exploring in future work, especially for further improvements in joint training of models that extract an object-centric representation, as the reviewer suggests. Our goal in this paper was to set up strong baselines using the most standard approaches, making them work in the complex problem setting posed by PHYRE. We hope our groundwork will allow the development of more sophisticated approaches inspired from SQUAIR, RNEM or SILOT on PHYRE.
>
> ### Modeling choices
>
> We apologize for any confusion.
>
>    1) __Encoders used:__ We mention the encoders used in Sec 4.1, first paragraph for each input type (object-based or pixel-based) . For object-based, it is a “tuple that contains object type (ball, stick, etc.), location, size, color, and orientation”. For pixel-based, “our image encoder is a ResNet-18 that is clipped at the res4 block”. The encoders are the same for all dynamics models considered for that input type. We will be releasing the full source code which will further clarify these details.
>
>    2) __Why is image represented as 7-channel?__ We do so to be comparable to prior work (Bakhtin et al. (2019)), where the image input to the network is represented as the 7-channel image. Thank you for pointing out the confusion, we have added a note in the text why this design decision was made.
>
>    3) __Input to the task solution model is the entire history:__ We do so to account for potentially poor performance of the dynamics model on certain tasks, as we want the model to at least perform comparably to a model without dynamics. So, for tasks where the dynamics model performs well, the task-solution model would learn to rely on the rolled out predictions; whereas for tasks where it performs poorly, it will learn to rely on original scenes to make a best guess and not trust the predictions. We have added this point to the text.
>
> ### Sampling actions
>
> We use the uniform random sampling to be consistent and comparable with prior work and existing baselines (Bakhtin et al. (2019)). As per our response to Reviewer 2, we have now also added analysis of our model’s performance at different number of actions being re-ranked to Appendix E. However, our approach can be further improved by using learned approaches to sampling actions, such as cross-entropy method (CEM), and could help further improve performance and reduce runtime.
>
>
> ### Other remarks
>
> Thank you for pointing those out. We have addressed all of those in the updated paper.

---

### Author Response · Authors · 2020-11-24
**Response to all Reviewers**

We thank all the reviewers for their time and insightful feedback. As the reviewers remarked, our work addresses a very challenging task (R2), providing the first forward prediction results on the PHYRE benchmark (R1) with thorough evaluation (R2, R3, R4, R5) resulting in interesting observations/analysis (R2, R3, R4, R5). We are glad they found the paper well written (R1, R3, R4) and an enjoyable read (R3, R4). We address a common concern in this common reply to all reviewers, and address all their specific concerns in individual responses. We have updated our paper to incorporate all the suggestions.

### Originality and novel insights

Several reviewers mention that our paper does not propose a novel technical solution. Indeed, our paper does not develop new methods. Instead, it performs an in-depth analysis of popular forward-prediction approaches on challenging physical-reasoning problems in the PHYRE benchmark. We believe analytical studies like ours are essential contributions to the community, as they provide important new scientific insights on the state of the field without being opinionated about any specific method. We emphasize that our work presents a number of novel contributions and insights over to prior work:

   1) We are the first to study  the effectiveness of forward prediction in physical-reasoning puzzle tasks in recent benchmarks like PHYRE or TOOLS. Making the models work at that large scale, to model all the different templates, is not trivial. We believe our paper and code release will be a major step forward in enabling researchers to experiment with their algorithms beyond block towers or multi-body systems.
   2) Our work achieves a new state-of-the-art performance on PHYRE, demonstrating the potential of forward prediction in physical reasoning.
   3) Our work is the first to highlight some of the limitations of modern forward-prediction methods for physical reasoning. In particular, our study shows that small deviations in forward predictions tend to negatively affect their performance on physical-reasoning tasks in a very significant way. This observation has substantial implications on the design of future physical-reasoning approaches based on forward predictions: indeed, these approaches need to be much more robust than simple search methods like the one we used in our study.
   4) Our work is the first to demonstrate that the success of recent forward-prediction methods may have caveats, in particular, that they may not generalize well across environments. Indeed, our study may be viewed as a call to arms to study generalization of forward-prediction methods much more rigorously.
   5) Our work provides new insights into the relative advantages and disadvantages of proprioceptive and pixel-based forward prediction methods. In particular, we find pixel-based methods to outperform proprioceptive ones in environments like PHYRE.
   6) We empirically show how pixel-based performance measures can be misleading, and that “visual realism”, as used in nearly all previous forward prediction work, may often not be the right metric to optimize when solving an actual downstream physical reasoning task. Benchmarks like PHYRE provide one such implicit evaluation methodology for physical reasoning, and our work establishes a new state of the art and provides extensive analysis.

We believe it is essential that ICLR provides a space for analytical studies like ours, as these scientific insights may impact the direction of our field in a long-lasting way by: (1) providing a realistic picture of the current state of the field without some of the noise and (2) providing clear guidance on which directions to pursue next on the field’s quest to solve the very difficult challenge that is physical reasoning.

---

### Decision · Program_Chairs · 2021-01-07
**Final Decision**

**Decision:**

Reject

**Comment:**

This paper evaluates several methods for physical prediction on the PHYRE benchmark, finding that while object-based methods (e.g. IN, Transformer) perform better in terms of predictive accuracy, pixel-based methods (e.g. STN, Deconv) perform better in terms of downstream task performance. The justification is that it is easier for the agent to evaluate good actions using an image-based representation rather than an object-based representation.

Pros:
- Important attempt to catalogue the current state of the field of physical reasoning
- Improved baselines on PHYRE

Cons:
- As pointed out by R5, there is a failure to evaluate any hybrid pixel-relational methods, such as OP3, R-NEM, C-SWM, etc. Given that the paper's main contribution is its assessment of the current state of the field (in the authors' own words: "providing a realistic picture of the current state of the field"), this seems like a major oversight to me.
- As pointed out by several reviewers, the analysis itself is somewhat limited. I don't see it as a problem that the paper does not propose any new methods, but in that case it needs to present a more thorough picture of why certain methods work better in some cases. For example, I share R1's concern that the Dec model performs worse than the identity function. Can you provide more detailed analysis demonstrating why the latent space is more useful? Can you demonstrate in what cases the object-based classifiers struggle, and why? Incorporating more careful hypotheses and ablations I think would help a lot in turning this into a much stronger paper.

I don't think it's a problem that the paper relies solely on 2D, fully-observed environments (many other papers on physical reasoning do this, so I think it's a reasonable choice), and I don't think it's a problem that the paper does not propose a new method. But I do find myself agreeing with the reviewers that the evaluations done within this context are insufficient. In the rebuttal, the authors emphasize the various conclusions stemming from the results (regarding the effect of model error, the extent of generalization, what "accuracy" means), but these conclusions are not that surprising (model error is a well-known problem in MBRL, deep models are notorious for their failure to achieve strong generalization, and the limitations of pixel accuracy have spawned whole research areas, such as contrastive and adversarial approaches). Again, I don't think the lack of surprising conclusions is itself an issue. But, the fact that the paper does not really make an attempt to explain any nuances or details regarding the conclusions makes it hard to draw a clear contribution from the paper; in that sense, I don't feel the paper really provides "clear guidance" as is argued in the rebuttal.

I do think this paper is very close to being acceptable, and could make a great submission to a future conference if the authors can spend a bit more time on (1) the baselines (i.e., incorporating hybrid models, and ensuring all methods pass basic gut checks) and (2) supporting their conclusions with more detailed analyses.